# Theoretical analysis reveals a role for RAF conformational autoinhibition in paradoxical activation

**Gaurav Mendiratta[1][†], Edward Stites[2,3]***

[1]Integrative Biology Laboratory, Salk Institute for Biological Studies, La Jolla, United States; [2]Department of Laboratory Medicine, Yale University, New Haven, United States; [3]Yale Cancer Center, Yale School of Medicine, New Haven, United States

**Abstract** RAF kinase inhibitors can, under certain conditions, increase RAF kinase signaling. This process, which is commonly referred to as 'paradoxical activation' (PA), is incompletely understood. We use mathematical and computational modeling to investigate PA and derive rigorous analytical expressions that illuminate the underlying mechanism of this complex phenomenon. We find that conformational autoinhibition modulation by a RAF inhibitor could be sufficient to create PA. We find that experimental RAF inhibitor drug dose–response data that characterize PA across different types of RAF inhibitors are best explained by a model that includes RAF inhibitor modulation of three properties: conformational autoinhibition, dimer affinity, and drug binding within the dimer (i.e., negative cooperativity). Overall, this work establishes conformational autoinhibition as a robust mechanism for RAF inhibitor-driven PA based solely on equilibrium dynamics of canonical interactions that comprise RAF signaling and inhibition.

**\*For correspondence:**
edward.stites@yale.edu

**Present address:** [†]Takeda Development Center America, San Diego, United States

**Competing interest:** The authors declare that no competing interests exist.

## Editor's evaluation

This important study uses mathematical modeling to demonstrate that conformational autoinhibition of the RAF kinase is an important feature of its paradoxical activation by pharmacological inhibitors. The provided theoretical analysis is highly compelling and increases the mechanistic understanding of how regulation by the N-terminal regulatory domains of RAF contributes to the paradoxical effect of the inhibitors. This article is poised to be of interest to biochemists, cell biologists, and cancer biologists, ultimately paving the way for improved RAF therapeutics.

## Introduction

Cancer is a disease that is characterized by a collection of shared phenotypes (*Hanahan and Weinberg, 2011*). Among these phenotypes is self-sufficiency in the pro-growth signals that aberrantly drive the continued proliferation of the cancerous cells. These excessive pro-growth signals may result from somatic mutations within a proliferation signaling pathway, and they could also result from excessive stimulation of transmembrane receptor proteins. Such pro-growth, proliferation, signals commonly pass through RAS GTPases (KRAS, NRAS, and HRAS) and RAF kinases (BRAF, CRAF, and ARAF) to initiate the RAF/MEK/ERK mitogen-activated protein kinase (MAPK) cascade.

Pharmacological agents that counteract these pro-growth signals have proven to be therapeutically beneficial in cancer. FDA-approved agents for upstream receptor tyrosine kinases that can drive proliferation through RAS/RAF signaling have proven effective for a variety of cancers. For example, EGFR inhibitors (both small molecule and therapeutic antibodies) have established benefit in lung, colorectal, and head and neck adenocarcinomas (*Ciardiello and Tortora, 2008*). Downstream from

**Figure 1.** Paradoxical activation (PA) of the RAS pathway and its potential mechanisms. (**A**) Schematic of the RAS/RAF/MEK/ERK signaling pathway and schematic of the PA concept. Whereas a RAF inhibitor can effectively inhibit RAF signaling to MEK in RAF mutant cancers, a RAF inhibitor can increase RAF signaling to MEK in cells without a RAF mutation (RAF wt). Gray triangles indicate directions of increasing RAF inhibitor (x-axis) and increasing RAF activity (y-axis) in the schematic. (**B**) Mechanisms that have been proposed to contribute to PA include RAF inhibitor-driven RAF dimer potentiation (DP), negative cooperativity (NC) for drug binding in trans within a RAF dimer once one protomer has bound drug, and the regulation of RAF kinase activity through conformational autoinhibition (CA).

receptor tyrosine kinases, the RAS GTPases have only recently become targetable, although for only one specific somatic mutation (KRAS G12C) (*Skoulidis et al., 2021*). Multiple drugs have been developed to inhibit the RAF kinases (*Bollag et al., 2010*; *Nakamura et al., 2013*; *Peng et al., 2015*; *Zhang et al., 2015*), and several of these agents have proven clinically valuable for RAF mutant melanoma and colorectal cancer.

The development of a drug that successfully inhibits its target protein is not sufficient to produce an effective drug. The drug must also avoid significant effects in non-diseased cells and tissues. This can be difficult, considering that the targeted protein could also be produced in the non-cancerous cells of the treated patient. For example, it was originally hoped that RAF inhibitors would be able to block the transmission of RAS signals (*Figure 1A*). Unexpectedly, RAF inhibitors were found to amplify RAS signals within the RAF wild-type context through a process that is now commonly referred to as 'paradoxical activation' (PA) (*Hatzivassiliou et al., 2010*; *Heidorn et al., 2010*; *Poulikakos et al., 2010*). PA is also associated with the development of drug-dependent squamous cell carcinomas in a subpopulation of those treated with RAF inhibitors (*Su et al., 2012*). Despite numerous studies, the mechanisms driving PA are still not fully understood (*Karoulia et al., 2017*; *Karoulia et al., 2016*; *Peng et al., 2015*).

A mathematical model of biochemical processes allows one to rigorously analyze what behaviors are possible for a given set of reaction mechanisms. Modeling can lead to mechanism-driven hypotheses

for further experimental testing (*Kholodenko, 2015*; *McFall et al., 2019*; *Rukhlenko et al., 2018*; *Stites et al., 2015*; *Stites et al., 2007*; *Wofsy et al., 1992*). Some of these mathematical models have proven useful for understanding mechanisms of anticancer agents. For example, we recently used a mathematical model to identify the mechanism (*McFall et al., 2019*; *McFall et al., 2020*; *McFall and Stites, 2021*) that explains why colorectal cancer patients with a KRAS G13D mutation benefit from treatment with EGFR inhibitors despite KRAS mutations more commonly causing resistance to EGFR inhibitors (*De Roock et al., 2010*). These studies are significant because a mechanistic explanation for this high-profile problem evaded determination through traditional experimental methods for approximately a decade. Thus, mathematical models can complement experimental approaches to provide new understandings to clinically relevant biological systems.

Mechanistically, two behaviors are commonly implicated as contributing to the phenomenon of PA (*Figure 1B*). First, some RAF inhibitors have been shown to result in an increased level of RAF dimerization (*Hatzivassiliou et al., 2010*; *Jin et al., 2017*; *Karoulia et al., 2016*; *Lavoie et al., 2013*). This drug-induced dimer potentiation (DP) is commonly thought of as manifesting in a higher affinity between RAF protomers when one (or both) are bound to a RAF inhibitor (*Kholodenko, 2015*). Second, many RAF inhibitors do not appear capable of binding to both protomers of a RAF dimer equally well (*Karoulia et al., 2017*; *Karoulia et al., 2016*; *Peng et al., 2015*). As only one protomer in a RAF dimer need be signaling competent for RAF signaling to propagate (*Heidorn et al., 2010*; *Hu et al., 2013*; *Yuan et al., 2018*), such negative cooperativity (NC) is posited to result in a reduced ability to fully inhibit RAF signaling. This NC is commonly thought of as manifesting in a reduced affinity for the binding of a second RAF inhibitor relative to the affinity with which the first RAF inhibitor bound (*Kholodenko, 2015*). Previous theoretical studies of PA have focused on these two mechanisms (*Kholodenko, 2015*; *Rukhlenko et al., 2018*).

Here, we report our mathematical analysis of RAF PA. We developed a mathematical model that describes key regulatory steps in RAF signaling. We developed the model to follow biochemical and thermodynamic principles, and we derive the behaviors that logically follow from the RAF regulatory mechanisms. Our model includes three different mechanisms that contribute to PA; two have received previous theoretical attention and one (RAF autoinhibitory conformational regulation) has not previously received a mathematical consideration. Our modeling reveals that, under certain conditions, RAF autoinhibitory conformational changes and their modulation by RAF inhibitor binding can be sufficient to drive PA. We fit our model to experimental data, and the fit model suggests links between the three types of RAF inhibitors and the three mechanisms that contribute to PA. Overall, this study suggests that all three mechanisms and their modulation by RAF inhibitor binding are necessary to explain PA.

## Results
### Analytical modeling of RAF autoinhibition

The regulation of RAF kinase activation is complex with multiple regulatory steps (*Hu et al., 2013*; *Karoulia et al., 2016*; *Lavoie and Therrien, 2015*; *Wellbrock et al., 2004*, *Brummer and McInnes, 2020*), and several of these processes have been described to play a role in PA (*Cho et al., 2012*; *Heidorn et al., 2010*; *Holderfield et al., 2013*; *Kondo et al., 2019*; *McKay et al., 2011*). Conformational changes of the RAF monomer contribute significantly to RAF kinase activation (*Jin et al., 2017*; *Lavoie and Therrien, 2015*; *Martinez Fiesco et al., 2022*; *Wellbrock et al., 2004*; *Zhang et al., 2021*). In the 'autoinhibited' form, associations between its N-terminus and its kinase domain maintain RAF in an inactive form that does not dimerize (*Cutler et al., 1998*; *Lavoie and Therrien, 2015*). In the 'non-autoinhibited' form, the kinase domain is no longer occluded, and other regulatory mechanisms contribute to full RAF kinase activation, such as kinase domain conformational changes and dimerization (*Lavoie and Therrien, 2015*; *Lavoie et al., 2013*). Recent experimental work reports that RAF inhibitors tend to promote a net transition to the non-autoinhibited conformation that is bound to RAS-GTP (*Jin et al., 2017*; *Karoulia et al., 2017*). It has previously been suggested that this biasing to the non-autoinhibited state may contribute to PA (*Figure 1B*; *Jin et al., 2017*). However, conformational autoinhibition (CA) has not previously received a theoretical analysis to evaluate whether, and how, it may contribute to PA.

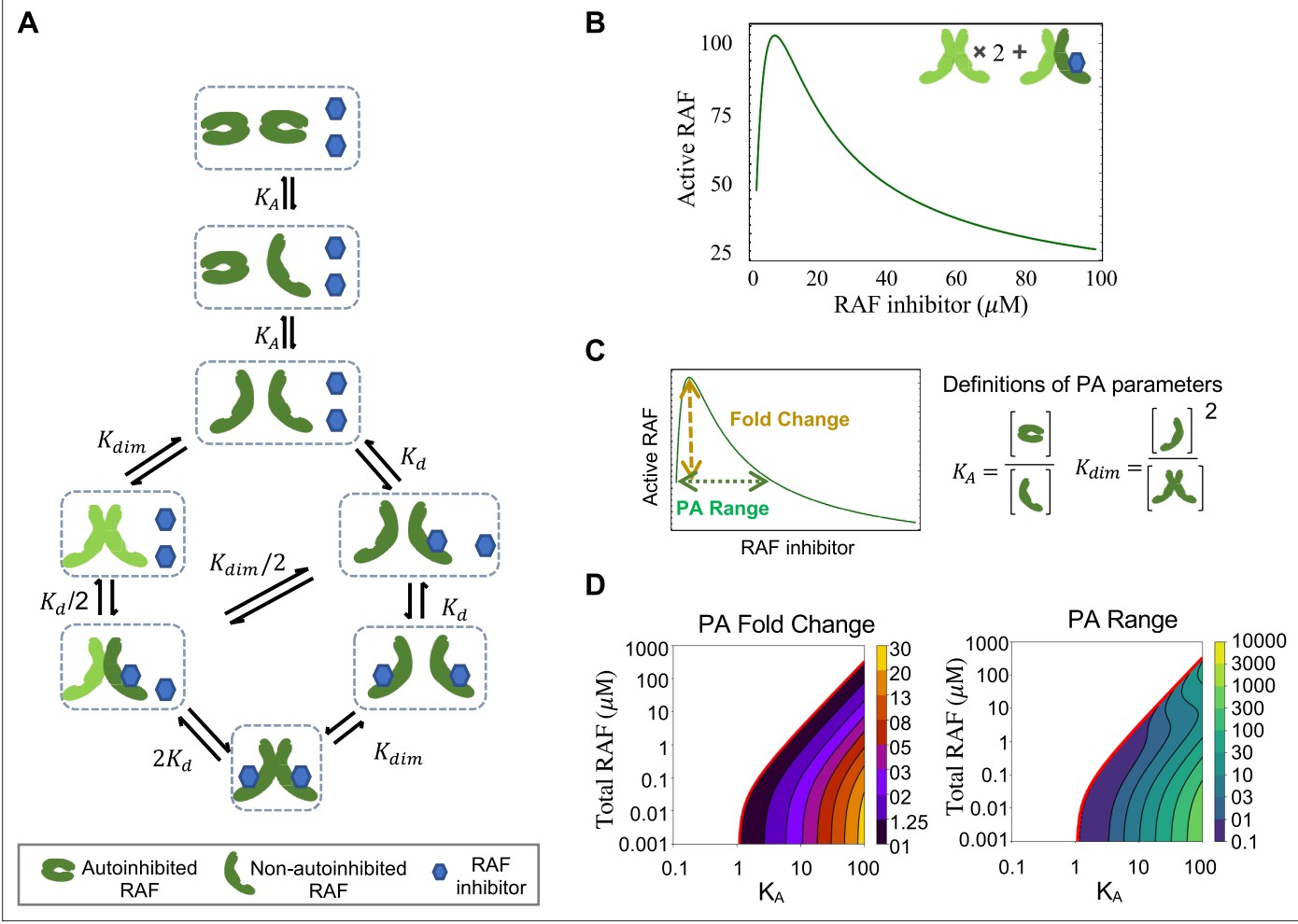

**Figure 2.** RAF autoinhibition is a mechanism that can produce paradoxical activation (PA). (**A**) Schematic of the RAF autoinhibition and dimerization model. RAF autoinhibition and the stabilization of the non-autoinhibited, dimerization, and signaling competent form of RAF by inhibitor is our first focus as a mechanism that may create PA. (**B**) Representative plot demonstrating that this model is sufficient to generate PA. Plotted quantity is the number of active RAF protomers (within a dimer, not bound to drug) normalized to the maximum as a function of RAF inhibitor abundance. (**C**) Schematic definitions of PA fold change, PA range, $K_A$, and $K_{dim}$. (**D**) Predicted PA fold change and range presented as a function of two key parameters of the autoinhibition model ($K_A$ and [RAF]). Panels (**B**) and (**D**) are numerical examples with specific parameter values of otherwise global, analytic results of the conformational autoinhibition (CA) model that are shown in *Table 1*.

The online version of this article includes the following figure supplement(s) for figure 2:

**Figure supplement 1.** Predicted paradoxical activation (PA) fold change and PA range as functions of the RAF inhibitor and RAF dimerization dissociation constants.

**Figure supplement 2.** Visualization of how paradoxical activation (PA) arises from conformational autoinhibition with inhibitor stabilization of the open from.

We therefore developed a series of mathematical models that include CA of RAF, NC within a RAF dimer, and RAF DP. We systematically studied different subsets of the full mechanism to investigate how each contributes to PA. For each sub-model, we derived analytic equations that provide the equilibrium solution for the system. One benefit of an analytic expression is that it can be utilized to determine the generality of a result and/or to identify specific criteria required for a given behavior to occur. Conclusions from most of our models are presented in theorem-proof form in Appendix 1 and are thereby devoid of any sensitivities arising from specific parameter choices. We refer to these conclusions as arising from 'analytical modeling.' We complement our analysis with 'numerical modeling' by using specific parameter values to create representative figures that can illuminate some of the predictions for the modeled network motif.

**Table 1.** Analytic conclusions in equation form for the different DP, NC, CA, and unified mechanisms models described in our work. Predicted expressions for active RAF at baseline (no drug), active RAF, total RAF dimers, and analytic conditions for PA to occur are shown in the rows for each of the mechanism associated models. The columns show models corresponding to the PA mechanisms of drug-modulated conformational autoinhibition (CA model), drug-induced dimerization potentiation mechanism (DP model), negative cooperativity toward second drug binding (NC model), and a model that combines all these mechanisms (unified model). The lower part of the table is a key that presents the abbreviated expressions, which allow presentation into similar functional forms for corresponding model results.

| Model type | Conformational autoinhibition (CA) | Dimer potentiation (DP) | Negative cooperativity (NC) | Unified model |
|---|---|---|---|---|
| Core model assumptions | 1. Monomeric RAF autoinhibits<br>2. Non-autoinhibited RAF can dimerize and bind drug<br>3. RAF dimers signal downstream | 1. RAF can dimerize and bind drug<br>2. Drug binding reduces dissociation constant of RAF dimers.<br>3. RAF dimers signal downstream | 1. RAF can dimerize and bind drug<br>2. Singly bound RAF dimers resist drug binding at second site<br>3. RAF dimers signal downstream | 1. CA model assumptions<br>2. DP model assumptions<br>3. NC model assumptions |
| Baseline active RAF (relative to total) | $\dfrac{\left(-1+\sqrt{(1+E3)}\right)^2}{E3}$ | $\dfrac{\left(-1+\sqrt{(1+8\,RAF_{rel})}\right)^2}{8RAF_{rel}}$ | $\dfrac{\left(-1+\sqrt{(1+8\,RAF_{rel})}\right)^2}{8RAF_{rel}}$ | $\dfrac{\left(-1+\sqrt{(1+E3)}\right)^2}{E3}$ |
| Active RAF (relative to total) | $\dfrac{\left(E1-\sqrt{E1^2+E2c}\right)^2}{E2c\left(1+d_{rel}\right)}$ | $\dfrac{\left(E1d-\sqrt{E1d^2+E2d}\right)^2 f\left(E1d+(-1+f)f\right)}{E2d\left(E1d^2+(-1+f)f^2\right)}$ | $\dfrac{E1n\left(E1n-\sqrt{E1n^2+E2n}\right)^2 g^2}{E2n\left(g+2gd_{rel}+d_{rel}^2\right)}$ | $\dfrac{\left(E1-\sqrt{E1^2+E2u}\right)^2 g\left(f+d_{rel}\right)}{E2u\left(fg+2gd_{rel}+d_{rel}^2\right)}$ |
| Total RAF dimers (relative to active RAF) | $\tfrac{1}{2}\left(1+d_{rel}\right)$ | $\dfrac{f+d_{rel}\left(2+d_{rel}\right)}{2\left(f+d_{rel}\right)}$ | $\dfrac{g}{2g\left(1+d_{rel}\right)}+d_{rel}$ | $\dfrac{fg+d_{rel}\left(2g+d_{rel}\right)}{2g\left(f+d_{rel}\right)}$ |
| RAF PA conditions | $RAF_{rel}<\tfrac{1}{8}\left(1+3K_A\right)\times\left(K_A-1\right)$<br>(** necessary and sufficient) | $d_{rel}<1-2f$<br>$8\times RAFrel<\left(3-8f+4f2\right)$<br>$f<1/2$<br>(* sufficient) | No PA possible | $g\geq 1$<br>$2f\geq 1+K_A$<br>$8\times RAF_{rel}<4f^2-8f\left(1+K_A\right)+3\left(1+K_A\right)^2$<br>(*sufficient) |
| Key | $E1d=\left(drel+1\right)f$<br><br>$E2d=8f\times RAF_{rel}\left(f+2d_{rel}+d_{rel}^2\right)$<br><br>$E1n=\left(d_{rel}+1\right)$<br><br>$E2n=8RAF_{rel}\left(g+2gd_{rel}+d_{rel}^2\right)$<br><br>*Sufficient: conditions that allow PA | | $E1=\left(d_{rel}+1+K_A\right)$<br><br>$E2c=8RAF_{rel}\left(1+2d_{rel}+d_{rel}^2\right)$<br><br>$E3=\left(8RAF_{rel}\right)/\left(1+K_A\right)^2$<br><br>$E2u=8RAF_{rel}\left(fg+2gd_{rel}+d_{rel}^2\right)/\left(fg\right)$<br><br>**Necessary and sufficient: *The only* conditions that allow PA. | |

Our first mathematical model focuses on RAF CA and RAF dimerization (*Figure 2A*). The model allows RAF to adopt two different conformations: one is autoinhibited and can neither dimerize nor bind drug, and the other is non-autoinhibited and can bind drug and/or dimerize (*Lavoie et al., 2013*). Drug-bound RAF is assumed to only be able to transition back to an autoinhibited state only after the bound drug has dissociated. Within the model, wild-type RAF is implicitly assumed to be activated by RAS-GTP as binding to RAS-GTP is an essential step to wild-type RAF activation (*Lavoie and Therrien, 2015*). We define active RAF as the RAF protomers that are not bound to a drug and are part of a RAF dimer.

The mechanism shown in *Figure 2A* and its conversion to a set of equations that provide the system's equilibrium solutions is detailed in Appendix 1. We use the principle of detailed balance (*Kholodenko, 2015*; *Wofsy et al., 1992*) and total protein and drug conservation equations to derive closed-form, analytic expressions for the steady-state solution for this system (*Table 1*, with derivations in Appendix 1). We focus on steady-state solutions because PA is a steady-state phenomenon, as is reflected in the long-term outgrowth of secondary tumors with RAS mutations from patients treated with a RAF inhibitor (*Su et al., 2012*).

## Paradoxical activation is a robust outcome of conformational autoinhibition

One benefit of an analytic expression is that it can be utilized to determine the generality of a result and/or to identify specific criteria required for a given behavior to occur. Along these lines, we investigated whether and under what conditions the mechanisms in our CA base model are sufficient to create PA. We can demonstrate analytically that the presence of both CA and stabilization of the active form by RAF inhibitors is sufficient to create PA for some, but not all, parameter values (*Figure 2B*, *Table 1*, Appendix 1). Additionally, our analysis yields algebraic expressions that define

the conditions necessary for PA to occur by this mechanism (*Table 1*). One condition we find is that RAF states must favor the autoinhibited form of RAF at equilibrium and in the absence of a drug for PA to occur. In other words, PA does not always happen when CA is present. By substituting biophysical parameter values into these expressions, we can predict PA for a wide range of RAF concentration values and visualize the range of system parameters for which CA promotes PA (*Figure 2C and D*, *Figure 2—figure supplement 1*). In our investigations, we also analytically show that PA is not generated by simpler mechanisms such as one without the autoinhibitory conformational changes and one without dimerization (Appendix 1).

## Paradoxical activation reflects a shifting balance of signaling complexes

To communicate how PA arises from this mechanism, we describe the proportion of RAF in each of its possible states: (i) autoinhibited RAF monomer, (ii) non-autoinhibited RAF monomer that is not bound to drug, (iii) non-autoinhibited monomer that is bound to drug, (iv) RAF dimer with no drug bound, (v) RAF dimer with one of two kinase domains bound to drug, and (vi) RAF dimer with both kinase domains bound to drug. We considered the total amount of kinase activity to be the number of RAF protomers within a dimer that are not bound to drug (*Figure 2—figure supplement 2*).

Before a drug is given, a significant fraction of RAF is autoinhibited and there are low levels of non-autoinhibited RAF and RAF dimers. As RAF inhibitor levels increase, the level of autoinhibited RAF progressively declines. Non-autoinhibited RAF distributes between drug-bound monomeric and dimeric forms while the unbound monomeric form maintains equilibrium with the autoinhibited RAF. The increased quantity of RAF dimers reflects the increased pool of RAF proteins that are non-autoinhibited and therefore capable of dimerization. This results in a drug-dependent increase in RAF dimers bound to drug in one site and thereby the increase in total RAF kinase activation that accounts for PA. The quantity of drug-bound RAF monomer and doubly drug-bound RAF dimer progressively increases to saturation as a function of the drug amount, resulting in the eventual reduction in RAF kinase activity that is associated with PA dose–responses.

## Evaluation of distinct mechanisms that promote paradoxical activation

There is evidence that NC and RAF inhibitor-driven DP also contribute to PA. Previous theoretical work on PA has focused on the contribution of DP and NC (*Kholodenko, 2015*; *Rukhlenko et al., 2018*). We were curious how CA might interact with these other processes that contribute to PA. We developed a mathematical model that combined the mechanisms studied in these previous studies with our model of CA; the complete mechanism that is modeled is shown in *Figure 3A*, and we adapt the same nomenclature for parameters (*Kholodenko, 2015*). Analysis of our model with CA, NC, and inhibitor-driven DP finds that these three mechanisms are mathematically distinct, in that one cannot translate or rescale variables to interchange between the mechanisms (*Table 1*). Further, each mechanism brings a qualitatively distinct feature to the dose–response of active RAF protomers (*Figure 3—figure supplement 1A and B*) and widely diverging capacity to potentiate the fold change in PA (*Figure 3—figure supplement 1A*) and the width of PA (*Figure 3—figure supplement 1B*). Because of these differences, we reasoned that a mathematical model that includes these biochemically, mathematically, and qualitatively distinct mechanisms may enable the underlying mechanisms of PA to be inferred, as in an inverse problem, through computationally fitting our model to data that characterizes the PA response of RAF inhibitors.

We developed a computational process for obtaining sets of parameters that fit our model to published experimental data (*Karoulia et al., 2016*). Our approach finds multiple parameter sets for the same drug that are within 10% of the error for the parameter set with the least error. These published data characterize the PA response of SK-MEL-2 cells to nine different RAF inhibitors, each at seven different concentrations of inhibitor (*Figure 3B*). We reasoned that some parameters should be specific to the individual drugs (i.e., the dissociation constant for the drug binding to RAF, and the drug-induced DP and NC parameters) and that some parameters should be the same no matter which drug was used in the experiment (i.e., the abundance of RAF, the equilibrium constant of RAF dimerization when no drug is bound, and the CA equilibrium constant). We set values for two key parameters, RAF abundance and RAF dimerization equilibrium constant, based on estimates and do not fit them. A table listing the best-fit parameters of each of the sub-models is provided in *Supplementary file 1*. Of note, we have 1 global parameter (the autoinhibition equilibrium constant) and 27

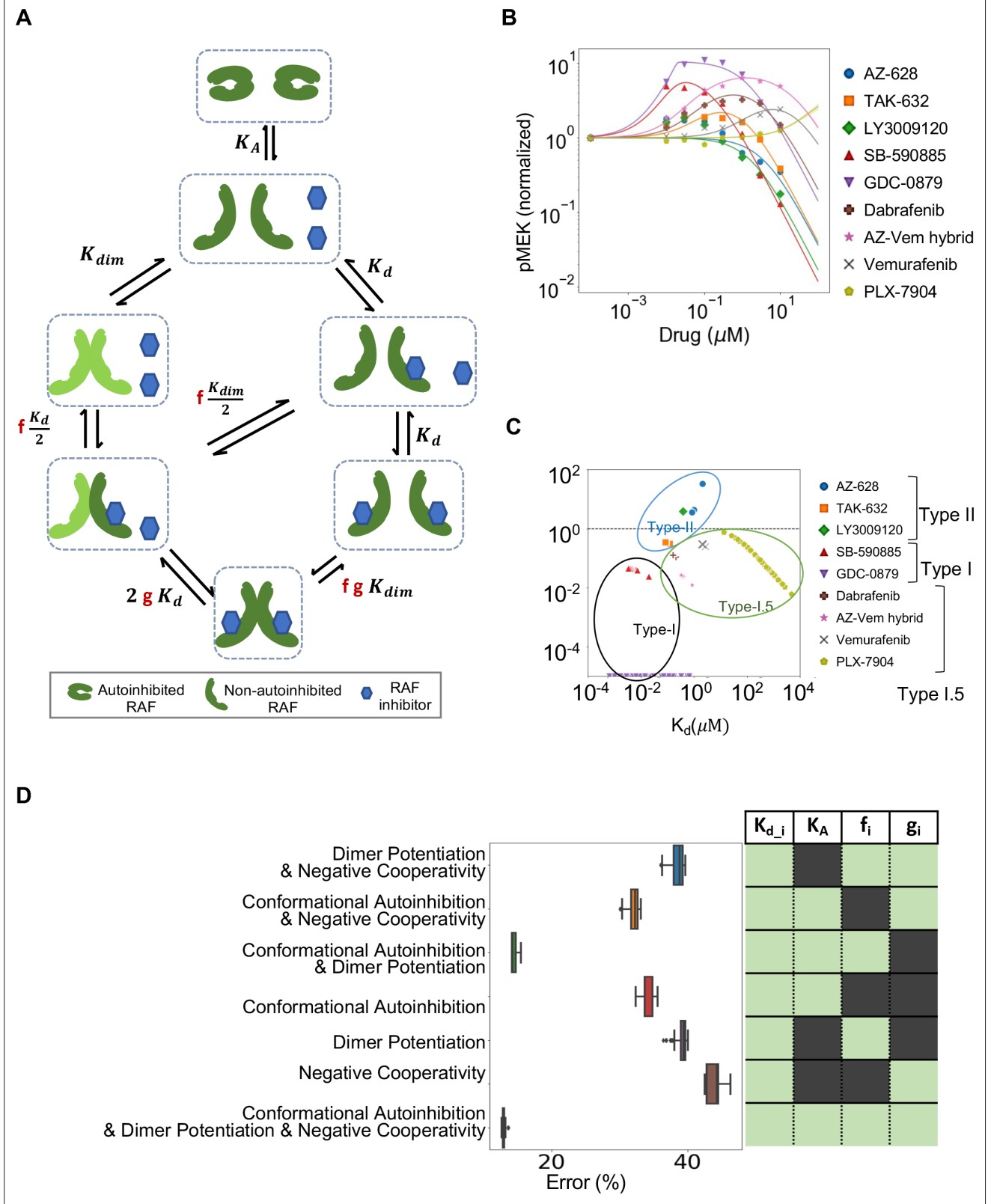

**Figure 3.** Estimated contribution of the three major mechanisms for promoting paradoxical activation (PA). (**A**) Schematic of the modeled mechanism of conformational autoinhibition extended to include dimer potentiation (characterized by parameter 'f') and negative cooperativity (characterized by parameter 'g'). (**B**) Plot of normalized dose–response curves for nine RAF inhibitors based on model fits to the available data (*Karoulia et al., 2016*). The solid lines represent mean values over N = 259 best fits for each of the 28 parameters varied. The standard deviation is highlighted in corresponding

*Figure 3 continued on next page*

*Figure 3 continued*

colored highlight. (**C**) The parameter 'f' and drug dissociation constant $K_d$ values from best-fit parameter sets of the unified model (N = 259) fit to nine RAF inhibitors are shown (best-fit $K_A$ = 2.914 ± 0.009). The outcomes of best fits for type I, II, and I.5 inhibitors are marked in black, blue, and green, ovals respectively. Dashed line at $f$ = 1 marks the absence of dimer potentiation mechanism. For each drug, we show all obtained best-fit parameter sets that were within 10% of best-fit metric. (**D**) Mean percentage error per input data for best-fit parameter sets in the unified model compared to models with one mechanism excluded and two mechanisms excluded. The panel on the right presents the parameters included in the corresponding model in light green (and those that are not in dark green). Subscript 'i' represents each of the nine drugs.

The online version of this article includes the following figure supplement(s) for figure 3:

**Figure supplement 1.** Characterizing dimer potentiation (DP), negative cooperativity (NC), and conformational autoinhibition mechanisms of paradoxical activation (PA).

**Figure supplement 2.** Best-fit dose–response curves for all models for each drug.

**Figure supplement 3.** Best-fit parameter values compared across different drugs.

drug-specific parameters. Our 28 total parameter estimates yield a model that matches the experimental data well (*Figure 3B*). Details of our approach for identifying model parameters are provided in the 'Materials and methods' section, and our code is provided in the supplement. We obtain similar quality overall fits if we do not pre-specify the RAF abundance and the RAF dimerization constant and add these two parameters to the other 28 parameters and then perform the same parameter estimation procedure (*Figure 3—figure supplement 1C–E*). We observed that some parameters could be constrained to a narrow region through this procedure, while other parameters could vary widely and still match the same data (*Figure 3—figure supplements 2 and 3*).

Intriguingly, when we considered the collection of best-fit parameters we found the different RAF inhibitors appeared to separate in a manner reflective of their known identity as type I, I.5, and II inhibitors (*Figure 3C*, *Supplementary file 1*). This is notable as only the RAF inhibitor dose–response data was examined to completely characterize the drug-dependent parameters, whereas the classification of kinase inhibitors is typically made by consideration of the structure of each kinase inhibitor complex (*Dar and Shokat, 2011*). We also plot alternative parameter estimates that are within 10% error of the considered parameter set with the least error, and this helps demonstrate the robustness of this observation. Thus, our model finds that the different mechanisms result in qualitatively distinct features and our fit parameters suggest that these differences enable RAF inhibitors to be grouped based on respective mechanisms of action.

## Conformational autoinhibition and dimer potentiation together best explain PA

We next evaluated how essential all three processes are to explain the PA data. We considered the accuracy of model fits for our full model with all three mechanisms, as well as models that only included one or two of these processes. We found that the unified model (i.e., the version that includes all three mechanisms) fits the data much better than any of these sub-models (*Figure 3D*). Only the simpler model that included both CA and DP (but not NC) could fit the data nearly as well.

With respect to NC, the addition of this process to the model did add an incrementally better fit. However, the data did not strongly constrain the value of 'g' that defines the extent of NC (*Figure 3D*, *Figure 3—figure supplements 2 and 3*). Additional experiments with a larger number of drug dose concentrations may in the future allow a better characterization of NC.

Altogether, we believe that our analysis provides strong support for CA being a critical factor to PA. Our sub-model analysis suggests that both CA and DP are necessary for the mathematical model to reproduce the experimental data with the smallest degree of error. Even though the NC term 'g' was not strongly constrained, and even though it had the smallest impact on the ability of the model to fit the experimental data, the model fits consistently had values of g that were much larger than 1 (*Supplementary file 1*), which is consistent with NC. Thus, our analysis suggests that all three processes contribute to the overall PA phenomenon.

## Discussion

From our studies, we conclude that CA, NC, and drug-induced increases in dimer affinity are required to explain the PA observed with the three major types of RAF inhibitors. Our mathematical analysis suggests that the inhibitor-modulated conformational regulation of RAF kinase activity combined with conformation-dependent dimerization is a critical mechanism that drives PA. Although inhibitor-modulated CA has been recognized as a possible contributor to the entire PA phenomenon (*Heidorn et al., 2010*; *Jin et al., 2017*), it has been underappreciated as a mechanism capable of driving PA. Supporting this assertion is that CA has neither been discussed as a motivation for the development of third-generation RAF inhibitors (*Nakamura et al., 2013*; *Peng et al., 2015*; *Zhang et al., 2015*) nor has it been included in recent mathematical analyses of PA and of RAF signaling (*Fröhlich et al., 2022*; *Kholodenko, 2015*; *Rukhlenko et al., 2018*; *Varga et al., 2017*). This is notable as both drug development and mathematical analyses pay considerable attention to conformations within the kinase domain (i.e., whether the alpha-C helix and DFG motif are in the 'in' or 'out' conformations) (*Kholodenko, 2015*; *Nakamura et al., 2013*; *Peng et al., 2015*; *Rukhlenko et al., 2018*; *Zhang et al., 2015*). That suggests that the concept of structural and conformational factors on PA is not foreign to drug developers and theoretical biologists, but that they may have limited their attention to the small changes in the kinase domain that have dominated the recent literature at the expense of also considering the large, autoinhibitory, conformational changes.

Additional processes have been claimed to also play a role in PA. For example, RAF protein phosphorylation changes (*Holderfield et al., 2013*), preferential binding of RAF inhibitor to the different RAF proteins and/or mutant RAF (*Bollag et al., 2010*), scaffold proteins (*McKay et al., 2011*), allosteric trans-activation (*Hu et al., 2013*), and RAS nanoclusters (*Cho et al., 2012*) may all further tune the response to inhibitor, including in drug-specific manners. 14-3-3 proteins can stabilize both RAF dimers (*Kondo et al., 2019*; *Park et al., 2019*; *Wilker and Yaffe, 2004*) and RAF CA (*Park et al., 2019*; *Wilker and Yaffe, 2004*), and it has been hypothesized that 14-3-3 stabilization of dimers plays roles in PA (*Kondo et al., 2019*; *Park et al., 2019*). Intuitively, stabilization of RAF CA by itself would result in a stronger PA and could be approximated as an increase in the $K_A$ term within our model. RAF stabilization of RAF dimers could be approximated as in increase in the $K_{dim}$ term within our model. Although any of these other processes could potentially modulate PA, we find that our mathematical model that includes CA, NC, and drug-induced increases in dimer affinity fits the experimental data well and that those fits reproduce orthogonal biochemical features of these inhibitors. Thus, we conclude that CA, NC, and drug-induced increases in dimer affinity are the three principal mechanisms underlying PA, and that the other processes listed above may provide additional modulation of the overall behavior.

It has previously been difficult to reconcile PA for type I.5 inhibitors, which are sometimes thought of as dimer breakers because they position the alpha-C helix in the 'out' position (in contrast to type I and II inhibitors). Studies with recombinant protein and analytic ultracentrifugation clearly found type I.5 inhibitors to predominantly be in the monomeric form (*Lavoie et al., 2013*). Within-cell assays have similarly found type I.5 inhibitors to promote dimerization less than other type I and II RAF inhibitors (*Hatzivassiliou et al., 2010*; *Peng et al., 2015*; *Thevakumaran et al., 2015*); however, RAF inhibitors still appeared to promote some dimerization in those in-cell assays. 14-3-3 binding proteins, which can help stabilize RAF dimers, may help explain this discrepancy (*Kondo et al., 2019*; *Liau et al., 2020*; *Park et al., 2019*). For example, by promoting the non-autoinhibited form, a type I.5 inhibitor-bound RAF monomer is more dimerization capable than an autoinhibited (and non-inhibitor-bound) RAF monomer. Even if the dimerization affinity is reduced compared to a non-autoinhibited and non-inhibitor-bound RAF monomer, 14-3-3 proteins may be able to bind and overcome the decrease in dimerization affinity. As our model does not explicitly include 14-3-3 proteins, this effect may contribute to our parameter estimation process finding an elevated binding affinity for type I.5-bound RAF monomers.

Although NC has been difficult to precisely measure experimentally, it has widely been assumed to be present to help explain the paradoxical activation caused by type I.5 inhibitors that do not promote dimerization as strongly as other RAF inhibitors. Our best-fit parameters did tend to have g values that were larger than 1, indicating that the model fit best when there was some NC. This could suggest that NC is more abundant than widely believed. Alternatively, the model without NC was able to fit the data nearly as well as the full model that included NC (i.e., *Figure 3D*). This may suggest

that other processes not included in the model may be modulating paradoxical activation and the g parameter, as the only other term in the model, is contributing to the model's ability to account for these otherwise not included effects.

We found parameter sets that reproduced available, published, data in order to test our model and investigate the potential for it to help illuminate aspects of PA. The best-fit parameter sets further support a role for CA and its modulation by RAF inhibitors in PA. However, it is also important not to read too deeply into the fits. For example, the data for the type II inhibitors AZ-628, LY3009120, and TAK-632 had small total fold change PA magnitudes, and our fits for them have even less PA. We anticipate that the model-fitting approach would converge to increasingly accurate estimates for the parameters as the set of data being fit to expands. Additionally, quantitative experimental measurements of the parameters being fit should also cascade to impact other parameters and result in better estimates (*Gutenkunst et al., 2007*).

Experimental testing of model predictions is an important next step. One possible approach for experimentally investigating the role of CA on PA would involve RAF proteins that are defective in CA. Of note, there are some RAF mutants in Noonan syndrome that are believed to be activating due to impaired regulation of CA. For example, stabilization of the autoinhibited form of CRAF involves the phosphorylation of serine at residue 259 followed by 14-3-3 binding to the phosphorylated serine. The S257L, S259F, P261A, and N262K CRAF mutations are impaired at binding to 14-3-3 and have severely impaired phosphorylation of serine 259 (*Kobayashi et al., 2010*; *Pandit et al., 2007*). Alternatively, mutants involving the autoinhibitory domain could be used (*Hartsough et al., 2018*; *Poulikakos et al., 2011*). In such experiments, comparisons of RAF inhibitor dose–responses between cells expressing one such mutant against control cells expressing the wild-type equivalent RAF protein would reveal whether there were changes in PA fold change and PA range (as defined in *Figure 2*). We would expect to see reduced PA fold change and reduced PA range for the CA-impaired mutants. We anticipate that PA range would be the more useful observable because nonlinearities in downstream signaling (including possible saturation of readouts like ERK phosphorylation) may make measurements of fold change less useful. As PA range is determined by when the signal passes back through the baseline level after the increase induced by a RAF inhibitor, it should be less impacted by nonlinear signal processing downstream from RAF. Whether the experiment utilizes Noonan syndrome RAF1 mutants or truncated RAF mutants, the presence of endogenous, wild-type BRAF, ARAF, and CRAF may make it difficult to observe differences in PA due to an introduced mutant, so such experiments may need to be performed in cells where the three endogenous RAF genes are knocked out or silenced.

Our analysis was motivated by RAF inhibitors and PA in RAS mutant cells treated with a RAF inhibitor. Our model, however, is generalizable to other systems that share the modeled features. We anticipate that PA will be observed for other proteins (i) that have a dynamic-equilibrium of conformations, (ii) where not all conformations can dimerize, and (iii) where drug binding the protein stabilizes one or more of the conformations that can dimerize. As dimerization and CA are both common features for kinase regulation (*Huse and Kuriyan, 2002*; *Lavoie et al., 2014*), it seems reasonable to hypothesize that PA will be observed for more kinases through modulation of the conformation and dimerization dynamic equilibrium.

## Materials and methods
### Mathematical models and analysis
We focus on steady-state levels of the different states in which RAF can exist, as portrayed in the diagrams for each model. Between any two states, an equilibrium relationship can be expressed as the ratio of abundances in the two states. Conservation of total protein quantities and zero value of total Gibbs free energy change at equilibrium both provide mechanisms to algebraically combine these expressions. We thereby derive expressions that relate the relative abundance of the RAF within its different monomeric and dimeric states. We perform algebraic manipulations and derive analytic solutions which we cross-check using Mathematica software v 12.0 (Wolfram Research). We perform numerical evaluations of these relationships and generate plots of these equations using Python packages, including numpy, scipy, and matplotlib.

Where applicable, parameters not varied in descriptive numerical plots are set as follows.

$K_A$ = 10.0, $K_d$ = 0.1 µM, $K_{dim}$ = 0.1 µM, [RAF] = 0.04 µM. We utilize mathematical models to clarify specific mechanisms that contribute to RAF activation and drug response. The models were driven by specific hypotheses that cannot be generalized to include all the features of RAF activation simultaneously due to limitations of available experimental data that does not cover all the required conditions and variations thereof. Moreover, our analytic approach allows a parameter-independent prediction for components of canonical RAF activation cycle that would not be accessible if we attempted to include every potential feature independent of a focused hypothesis. This is similar to differences in complexity between in vitro and in vivo systems – the former often forms a more versatile and precise method of establishing mechanisms while not including most of the complexities present in vivo, thereby providing a precise testing ground for focused hypotheses.

## Model fitting to dose–response data

We reasoned that some parameters (such as the abundance of the RAF kinases, their dimerization rate constant [$K_{dim}$], and the equilibrium constant for RAF conformational changes [$K_A$]) are intrinsic to a cell and should not vary between different RAF inhibitors. The values of RAF concentrations and dimerization constant are also effective parameters since all three types of RAF are implicitly included when we fit to in vitro data. In contrast, some properties will vary between RAF inhibitors. Specifically, these would be the affinity of a drug for binding to RAF ($K_d$), the drug-induced change in RAF dimer affinity, or "f" to use the nomenclature of *Kholodenko, 2015*, and NC, or "g" to again use the nomenclature of *Kholodenko, 2015*.

Numerical fitting for unified model of PA mechanisms was performed using SLSQP algorithm from scipy library in Python language. Sixty-three experimental dose–response points from the PDF publication (*Karoulia et al., 2016*) were quantified using LI-COR software (Image Studio V5.2) and used as raw data to fit 28 (*Figure 3*, *Supplementary file 1*) or all 30 parameters (*Figure 3—figure supplement 1C–E*, *Supplementary file 1*) in our model for the nine drugs included. A total of 1500 random initial states were chosen from a log-uniform search domain for each of the parameters. The boundaries of the fitting search were identified in the units of µM (where applicable) as follows: 'f,' [$10^{-5}$,100]; 'g,' [1,$10^4$]; '$K_A$,' [0.001,$10^2$]; '$K_d$,' [$10^{-4}$,$10^4$]; '$K_{dim}$,' [$10^{-4}$,$10^4$]; 'RAF,' [$10^{-4}$,$10^3$]. Best fits were defined as within 10% of the lowest value of the fitting metric. Several fitting algorithms and metrics were evaluated to finally identify SLSQP with a chi-square-like fit metric leading to convergence with higher likelihood across different initial conditions. This metric identified deviation across all data points relative to the model prediction as follows:

$$fitmetric = \frac{1}{Ndrugs * Ndoses} \sqrt{\sum_{i=0}^{Ndrugs} \sum_{j=0}^{Ndoses} \left( \frac{y_{i,j}^{predicted} - Y_{i,j}^{observed}}{Y_{i,j}^{observed}} \right)^2}$$

To compare how well each of the sub-models fit the data relative to one another, we optimized over the absolute error relative to the data as this metric has a straightforward interpretation of average proportionate deviation.

$$fitmetric = \frac{1}{Ndrugs * Ndoses} \sum_{i=0}^{Ndrugs} \sum_{j=0}^{Ndoses} |\frac{y_{i,j}^{predicted} - Y_{i,j}^{observed}}{Y_{i,j}^{observed}}|$$

In 30 parameter fits that varied all the parameters in our models, the value for RAF concentration was best fit to be 0.033 ± 0.005 µM for SKMEL2 cells (*Supplementary file 1*). In these fits, we also found that the dimerization equilibrium constant of RAF is monotonically correlated to autoinhibition constant $K_A$ creating a non-identifiability problem (*Figure 3—figure supplement 1E*). Hence, other than *Figure 3—figure supplement 1C–E*, the model fits were performed with a representative choice of RAF concentration fixed at 0.04 µM and RAF intrinsic equilibrium dimerization dissociation rate fixed at 0.1 µM. These parameters vary with cellular context, and the chosen values are within the range of values observed in the literature (*Fujioka et al., 2006*; *Lavoie et al., 2013*; *Sadaie et al., 2014*).

Mathematica and Python files that allow for the reproduction of both our analytic and numerical analyses respectively are provided as supplementary data files and are made public on GitHub link (https://github.com/GMendiratta/RAF-PA copy archived at *Mendiratta, 2023*).

## Code availability

All codes needed to reproduce the work presented in the article are presented in the supplementary code. The codes are also available on GitHub (https://github.com/GMendiratta/RAF-PA).

## Acknowledgements

We thank Tony Hunter, Geoff Wahl, Reuben Shaw, Bjorn Lillemeier, Dmitry Lyumkis, Joe Noel, Andrey Shaw, Rajasree Kalagiri, Leo Li, Michael Trogdon, Noah DeTal, Peter Carlip, and Melinda Tong for helpful conversations and feedback. We thank Amy Cao for help with illustrative graphics. This work was supported by a Salk Pioneer Fund Postdoctoral Scholar Award (GM); the National Institutes of Health NIH K22CA216318 (ECS), NIH DP2AT011327 (ECS), a Melanoma Research Alliance Young Investigator Award (ECS), the Joe W and Dorothy Dorsett Brown Foundation (ECS), and the Conrad Prebys Foundation (ECS).

## Additional information

### Funding

| Funder | Grant reference number | Author |
|---|---|---|
| National Institutes of Health | K22CA216318 | Edward Stites |
| National Institutes of Health | DP2AT011327 | Edward Stites |
| Melanoma Research Alliance | Young Investigator Award | Edward Stites |
| Joe W. and Dorothy Dorsett Brown Foundation | | Edward Stites |
| Salk Institute for Biological Studies | | Gaurav Mendiratta |
| Conrad Prebys Foundation | | Edward Stites |

The funders had no role in study design, data collection and interpretation, or the decision to submit the work for publication.

### Author contributions

Gaurav Mendiratta, Conceptualization, Data curation, Software, Formal analysis, Validation, Investigation, Visualization, Methodology, Writing – original draft; Edward Stites, Conceptualization, Formal analysis, Supervision, Funding acquisition, Investigation, Methodology, Writing – original draft

### Author ORCIDs

Gaurav Mendiratta ORCID https://orcid.org/0000-0001-5091-348X
Edward Stites ORCID https://orcid.org/0000-0002-3783-7336

### Decision letter and Author response

Decision letter https://doi.org/10.7554/eLife.82739.sa1
Author response https://doi.org/10.7554/eLife.82739.sa2

## Additional files

### Supplementary files

• MDAR checklist

• Source code 1. Zipped folder containing supplementary code that enables all results to be reproduced. Contents include a folder that includes Mathematica notebooks that derive the analytic results. Contents also include a folder that includes Python notebooks that reproduce all of the numerical results.

• Supplementary file 1. Table of best-fit estimates for mechanistic parameters within the unified mechanisms model and all sub-models. From top to bottom: best-fit parameters are shown for the DP + NC (blue), CA + NC (orange), CA + DP (green), CA (red), DP (violet), NC (brown), CA + NC + DP unified mechanisms model when 28 (black) or all 30 (gray) parameters are varied. Each of the models are fit to 63 input data points obtained from Figure 4E of *Ciardiello and Tortora, 2008*. Input pMEK data was normalized to total MEK and then for each of the nine drugs normalized to the respective drug-free (DMSO) condition to enable comparison with model predictions. Six parameters ($f$, $g$, $K_d$, $K_A$, $K_{dim}$, [RAF]) identify dose–response curve generated by each drug. Of these, three parameters ($f$, $g$, $K_d$) are specific to each drug, totaling to 27 parameters for the nine drugs measured. Three parameters are cell-context specific ($K_A$, $K_{dim}$, [RAF]). For all parameter sets other than unified model 30 parameters varied, [RAF] was set to 0.04 μM and $K_{dim}$ was set to 0.1 μM. All numbers shown as mean ± SD. Numerical simulations of the unified model are shown in *Figure 3— figure supplement 1*.

## Data availability

All data needed to evaluate the conclusions in the paper are present in the paper or the Supplementary Materials. All materials are available upon request from the corresponding author.

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

# Appendix 1

## Supplementary theory

The theory of ordinary differential equations (ODEs) was developed alongside calculus by Newton, Leibniz, Bernoulli, Riccati, and others in the 17th century. Nonlinear differential equations are extensively used in research including to model biological systems of neural signaling, protein interactions, and cardiac potentials; however, these are rife with singularities and numerical instabilities. Through this long history, differential equations that can be fully solved and are applicable to real-world conditions are quite rare. Hence, scientifically relevant, analytically accessible solutions are of considerable value to both progress of applied mathematics and quantitative sciences. In this work, we show such solutions under equilibrium conditions for a set of ODE models that describe the dimerization and drug binding of RAF proto-oncoproteins. Our work includes a series of mathematical models involving RAF dimerization, drug binding, RAF CA, and the interactions that affect these component processes. Each of these models is based on evidence from multiple sources in the literature that support the modeled RAF activation cycle and drug interactions. The value of such analytically solved models is both in their generalizability and in their global predictions. For example, in the first section, we will show that it is not possible to generate PA in a system where RAF monomer can dimerize and where drug binds equivalently well to monomeric and dimeric RAF proteins. This statement is derived independent of any parameter values that may be used – this is an 'analytic' result. Other examples in biological literature of similarly powerful analytic studies include Michaelis–Menten enzyme kinetics (*Michaelis and Menten, 1913*), negative feedback-based oscillatory circuits (*Glass and Mackey, 1988*; *Lewis, 2003*), and the Hodgkin–Huxley model (*Hodgkin and Huxley, 1952*; *Noble, 1962*).

The molecular interaction rates for RAF activation and drug binding all occur on a time scale of protein–protein and protein–ligand interactions (*Fujioka et al., 2006*; *Gillies et al., 2020*), which are orders of magnitude smaller than the time scales of paradoxical activation and proliferation phenotypes associated with MAPK pathway activation (*Hatzivassiliou et al., 2010*; *Heidorn et al., 2010*) or the time scale of clinical impact (*Gibney et al., 2013*). We therefore focus on steady-state levels of the different states in which RAF can exist within our models. At equilibrium, we derive expressions for the relative abundance of the RAF within its different monomeric and dimeric states. We perform algebraic manipulations and derive analytic solutions that we reproduce using Mathematica software. We solve the equilibrium state numerically where it is instructive as visual aid to the analytical results or where analytic results cannot be obtained globally.

Within Appendix 1, we first describe a model of RAF dimerization and drug binding, followed by a model of autoinhibition without dimerization, and then a model that combines these two components. A numerical illustration of the mathematical model used that displays the drug dose-dependent response of the different states of RAF is included as *Figure 2—figure supplement 2*, with narrative text in the main article under the section ' Paradoxical activation reflects a shifting balance of signaling complexes.' The NC and DP mechanisms utilized in unified mechanisms model have been discussed in detail in the literature (*Kholodenko, 2015*). We show the resulting expressions in the unified model in *Supplementary file 1* (and use these in *Figure 3* plots and fits) and the derivations thereof are included in Mathematica code present in the included supplementary code files. We provide a numerical illustration of the unified model that displays the drug dose-dependent response of the different states of RAF included as the last section of Appendix 1.

## The promotion of RAF dimerization by a drug does not induce PA by itself

We begin by constructing a model of kinase activation to dispel a possible misunderstanding of the behavior possible through RAF dimerization and drug binding alone. In the mechanism that we model (*Appendix 1—figure 1*), a kinase exists in the competent/non-autoinhibited form (denoted by capital *A*) where it may dimerize (*AA*) and bind with the drug in the monomeric (*Ad*) or dimer form (*AAd,AdAd*). This model resembles a similar, older model of EGFR activation and dimerization (*Wofsy et al., 1992*). We can assign an equilibrium constant to each of the reactions that is equal to the ratio of reactants to the products:

$$K = \frac{[Reactant_1]\,[Reactant_2]}{[Product]}$$

Therefore, the process of drug binding has an equilibrium rate constant $K_d$:

$$K_{A+d>Ad} = K_d = \frac{[A]\,[d]}{[Ad]}$$

The process of non-autoinhibited kinase dimerization corresponds to the equilibrium rate constant $K_{dim}$:

$$K_{A+A>AA} = K_{dim} = \frac{[A]\,[A]}{[AA]}$$

We derive closed-form, analytic steady-state solutions relating the above rate constants with protein and drug concentrations using the principle of detailed balance (**Kholodenko, 2015**; **Wofsy et al., 1992**). At equilibrium, the principle of detailed balance or microscopic reversibility ensures that all cyclical processes have a zero change in Gibbs free energy. Free energy with a constant temperature, pressure, and pH is proportional to the logarithm of the equilibrium rate constant. Hence, corresponding to the network described in **Appendix 1—figure 1**, we obtain the following relations:

$$\frac{\dfrac{[A]\,[A]}{[AA]}\dfrac{[AA]\,[d]}{[AAd]}\dfrac{[AAd]}{[Ad]\,[A]}\dfrac{[Ad]}{[A]\,[d]}}{\dfrac{K_{A+A>AA} \times K_{AA+d>AAd}}{K_{Ad+A>AAd} \times K_{A+d>Ad}}} = 1 \qquad = 1$$

and

$$\frac{\dfrac{[Ad]\,[A]}{[AAd]}\dfrac{[AAd]\,[d]}{[AdAd]}\dfrac{[AdAd]}{[Ad]\,[Ad]}\dfrac{[Ad]}{[A]\,[d]}}{\dfrac{K_{Ad+A>AAd} \times K_{AAd+d>AdAd}}{K_{Ad+Ad>AdAd} \times K_{A+d>Ad}}} = 1 \qquad = 1$$

Since there are two binding locations for the drug in a dimer ($AA + d > AAd$), the reaction is twice as likely, and the dissociation constant is halved $K_d/2$. From the detailed balance equation, we calculate that $K_{Ad+A>AAd} = K_{dim}/2$, hence connecting drug binding to the dimerization process via dynamical equilibrium relationships. In this first, simple, model, we assume that the drug does not induce any changes in the kinase conformation and therefore the dimerization of drug-bound kinase proceeds at the same rate as unbound kinase, $K_{Ad+Ad>AdAd} = K_{dim}$. We can now calculate the last unknown rate, $K_{AAd+d>AdAd} = 2K_d$. Thereby, only two rate constants determine the equilibrium state of this system.

To calculate the steady-state or equilibrium relationships, we begin with reactions that lead from a RAF protomer to a fully drug-bound RAF dimer:

$$\begin{aligned} A + d &\rightleftharpoons Ad \\ A + A &\rightleftharpoons AA \\ AA + d &\rightleftharpoons AAd \\ AAd + d &\rightleftharpoons AdAd \end{aligned}$$

The following expressions correspond to the rate constants for the above reactions:

$$
\begin{aligned}
[Ad] &= \frac{[A]\,[d]}{K_d} \\
[AA] &= \frac{[A]^2}{K_{dim}} \\
[AAd] &= 2\frac{[AA]\,[d]}{K_d} \\
&= 2\frac{[A]^2\,[d]}{K_{dim}K_d} \\
[AdAd] &= \frac{[AAd]\,[d]}{2K_d} \\
&= \frac{[A]^2\,[d]^2}{K_{dim}K_d^2}
\end{aligned}
$$

The total amount of drug and kinase is obtained by combining all states that are assumed unchanged over the short reaction times involved in the binding and unbinding molecular reactions, and are defined as below:

$$
\begin{aligned}
[A] + [Ad] + 2\left([AA] + [AAd] + [AdAd]\right) &= RAF \\
[d] + [Ad] + [AAd] + 2[AdAd] &= Drug
\end{aligned}
$$

Replacing the equilibrium concentrations into the expression for total kinase concentration, we first solve for the total number of unbound RAF protomers:

$$
\frac{[A]}{RAF} = \frac{\sqrt{1 + 8RAF_{rel}} - 1}{4\left(1 + d_{rel}\right) RAF_{rel}}
$$

Here, $RAF_{rel} = RAF/K_{dim}$ and $d_{rel} = [d]/K_d$. The total quantity of active protomers is a sum of the protomers in the kinase dimers and the dimers bound to one drug, $(2[AA] + [AAd])$. Substituting the equilibrium concentrations, the concentration of active kinase at equilibrium is obtained:

$$
2[AA] + [AAd] = 2\frac{[A]^2}{K_{dim}} + \frac{[A]^2\,[d]}{K_{dim}K_d}
$$

Utilizing the previously calculated equilibrium concentration of non-autoinhibited monomer kinase, we can reduce the above expression to a function of the dimensionless ratios $RAF_{rel}$ and $d_{rel}$:

$$
\frac{2[AA] + [AAd]}{RAF} = \frac{\left(1 - \sqrt{1 + 8RAF_{rel}}\right)^2}{8\left(1 + d_{rel}\right) RAF_{rel}}
$$

It is clear that the above expression for active kinase protomers is a monotonically reducing function of the unbound drug concentration since that variable only occurs in the denominator. The second component required to analytically establish the absence of paradoxical activation in this model is to prove that the unbound drug is monotonically and proportionally related to total drug concentration. We do this by substituting the equilibrium relations, which provides the total drug amount:

$$
Drug = [d] + \frac{[A]\,[d]}{K_d} + \frac{2[d]\,[A]^2\left([d] + K_d\right)}{K_d^2 K_{dim}}
$$

Next, we substitute the equilibrium kinase monomer concentration [A] in the above expression to solve for equilibrium unbound drug concentration [d].

$$
[d] = \frac{1}{2}\left(\zeta_0 + \sqrt{4K_d Drug + \zeta_0^2}\right)
$$

Here, $\zeta_0 = Drug - RAF - K_d$. Therefore, the relationship between total and unbound drug concentrations is monotonic and directly related.

We therefore show that active RAF is an inverse function of drug concentration. *PA is not possible while considering only simple dimerization and drug binding processes of RAF protein.* This point may not have been sufficiently stressed in the literature, leading to confusion about the added dimers

are induced by the drug causing PA by themselves. These new, partly drug-bound, dimers will never create additional RAF signaling protomers unless there was more to the RAF activation processes than dimerization and drug binding. For example, DP, where the affinity for RAF dimerization is higher if one (or both) RAF protomers is bound to a RAF inhibitor, has previously been demonstrated as an addition to the above modeled mechanism that could result in a PA response to RAF inhibitor (*Kholodenko, 2015*).

## Autoinhibited proteins active as monomers will not display PA from drug-induced stabilization of the non-autoinhibited form alone

Extensive research has searched for explanations of PA that go beyond DP. These mechanisms include drug NC, RAS oligomerization, downstream negative feedback release, and more. One of these mechanisms, based on RAF CA, is particularly interesting because it does not involve any new, drug-induced, structural conformations of the RAF protein. With a simple extension to our modeled network, we identify the conditions under which this phenomenon successfully provides a mechanism for PA in response to inhibition of dimerizing kinases.

We first model drug-mediated stabilization of the active form of a conformation autoinhibition-regulated kinase for a kinase that does not dimerize and is fully active as a monomer. To study this, we create a model where the kinase may autoinhibit (denoted by '*a*') to become reversibly inactive and where a drug will only bind the non-autoinhibited form (*Appendix 1—figure 2*). The following equilibria contribute to this network:

$$A + d \quad \rightleftharpoons Ad$$
$$A \quad \rightleftharpoons a$$

with corresponding equilibrium dissociation constants defined as follows:

$$K_d = \frac{[A][d]}{[Ad]}$$
$$K_A = \frac{[a]}{[A]}$$

The total drug and kinase amounts are calculated as below:

$$[a] + [A] + [Ad] = RAF$$
$$[d] + [Ad] = Drug$$

We solve the above four equations for the total non-autoinhibited protomers ([A]) as a function of total concentrations and equilibrium dissociation constants:

$$A = \frac{\sqrt{C_1^2 + 4(1 + K_A) K_d RAF} + C_1}{2(1 + K_A)}$$

where $C_1 = RAF - Drug - K_d(1 + K_A)$. The function monotonically reduces to zero as the total drug increases to a large value. Therefore, *a kinase that can signal downstream as a monomer does not show anomalous activation upon drug inhibition even though there is a reservoir available to shift the equilibrium to activated kinase.*

## The combination of conformational autoinhibition and stabilization of the active form by a drug can generate PA

Conformational changes of the RAF monomer contribute significantly to the regulation of RAF kinase activity (*Lavoie and Therrien, 2015*; *Wellbrock et al., 2004*). In the 'autoinhibited' form, associations between its N-terminus and its kinase domain maintain RAF in an inactive form that does not dimerize (*Cutler et al., 1998*; *Lavoie and Therrien, 2015*). In the 'non-autoinhibited' form, the kinase domain is no longer occluded, and other regulatory mechanisms contribute to full RAF kinase activation, such as kinase domain conformational changes and dimerization (*Lavoie et al., 2013*). Recent experimental work reports that RAF inhibitors tend to promote a net transition to the non-autoinhibited conformation that binds to RAS-GTP (*Jin et al., 2017*; *Karoulia et al., 2017*). It

has previously been suggested that this biasing to the non-autoinhibited state may contribute to PA (*Jin et al., 2017*).

To study the conditions under which the stabilization of RAF in its non-autoinhibited state by RAF inhibitors may be sufficient to generate PA, we create a mathematical model that includes RAF CA, RAF dimerization, and RAF inhibitor (*Appendix 1—figure 3*). The model allows RAF to adopt two different conformations: one is autoinhibited and can neither dimerize nor bind drug, and the other is non-autoinhibited and can bind drug and/or dimerize (*Lavoie et al., 2013*). Drug-bound RAF is assumed to only be able to transition back to an autoinhibited state after any bound drug has dissociated. Within the model, wild-type RAF is implicitly assumed to be activated by RAS-GTP as binding to RAS-GTP is an essential step to wild-type RAF activation (*Lavoie and Therrien, 2015*). We define active RAF as the RAF protomers that are not bound to a drug and are part of a RAF dimer. The following expressions identify the different interactions included in this network:

$$
\begin{aligned}
a &\rightleftharpoons A \\
A + d &\rightleftharpoons Ad \\
A + A &\rightleftharpoons AA \\
AA + d &\rightleftharpoons AAd \\
A + Ad &\rightleftharpoons AAd \\
AAd + d &\rightleftharpoons AdAd \\
Ad + Ad &\rightleftharpoons AdAd
\end{aligned}
$$

The processes in this model include autoinhibition, which, for simplicity, is modeled as a first-order transition from autoinhibited state $a$ to non-autoinhibited state $A$ with a rate constant $K_A$:

$$
K_{a \to A} = K_A = \frac{[a]}{[A]}
$$

The equilibrium equations defined in the previous section apply in the base model with the addition of the CA dynamical equilibrium represented in above equation. Over the reaction periods, we assume that the total RAF and drug concentrations remain unchanged, leading to the conservation equations below:

$$
\begin{aligned}
RAF &= a + A + Ad + 2\left(AA + AAd + AdAd\right) \\
Drug &= d + Ad + AAd + 2AdAd
\end{aligned}
$$

Following the analysis described in the first section, we solve the RAF conservation equation to find the concentration of unbound non-autoinhibited RAF:

$$
\frac{[A]}{RAF} = \frac{K_d K_{dim}}{4\left(d + K_d\right)^2} \times \sqrt{\left([d] + K_d + K_A K_d\right)^2 + 8\left([d] + K_d\right)^2 \times \frac{RAF}{K_{dim}}} - \left([d] + K_d + K_A K_d\right)
$$

Note that there is no explicit dependence on the drug binding rate $K_d$ and only on the ratio of the free drug concentration to $K_d$ appears explicitly ($d_{rel} = [d]/K_d$). Similarly, dimerization rate constant $K_{dim}$ can be combined with the total amount of RAF protein $RAF_{rel} = [RAF]/K_{dim}$. This reduces the model from four varying parameters to two effective parameters, which govern all predictions. The total active kinase is the sum of drug-unbound RAF protomers in a drug-unbound RAF dimer and partly drug-bound RAF dimer.

$$
\begin{aligned}
ActiveRAF &= 2\left[AA\right] + \left[AAd\right] \\
&= \frac{2\left[A\right]^2}{K_{dim}} + \frac{2\left[A\right]^2\left[d\right]}{K_{dim} K_d}
\end{aligned}
$$

Substituting the above equation and the equilibrium conditions, we obtain the active RAF proportion shown in the first column of *Supplementary file 1*. This function is not monotonic, and the behavior can be defined by evaluating the zeroes of the derivatives of the function. The first derivative of the function representing active RAF protein simplifies to the following expression:

$$\frac{d\left(ActiveRAF\right)}{d\left(d_{rel}\right)} = \frac{\left(E1 - \sqrt{E1^2 + E2}\right)}{E2\left(d_{rel} + 1\right)^2} \times \left(2\left(d_{rel} + 1\right)\left(1 - \frac{\frac{E2}{d_{rel} + 1} + E1}{\sqrt{E1^2 + E2}}\right) + 3\left(\sqrt{E1^2 + E2} - E1\right)\right)$$

Here, $E1 = 1 + K_A + d_{rel}$ and $E2 = 8\left(1 + d_{rel}\right)^2 RAF_{rel}$ . Maximal paradoxical activation, if it exists, occurs when the first derivative of active RAF relative to the drug concentration is zero. We calculate the values of $d_{rel}$, which satisfy this condition and find that only one solution can potentially be real and positive:

$$d_{relmaxFC} = \frac{1}{8RAF_{rel} + 1}\left(2K_A\sqrt{\left(6RAF_{rel} + 1\right)} - 1 - 8RAF_{rel} - K_A\right)$$

To additionally establish that this point of inflection is a maximum, we need to establish conditions under which the second derivative of the active RAF equation relative to $d_{rel}$ is negative. At the drug concentration in the above equation, the second derivative of active RAF simplifies to the following expression:

$$\frac{d^2\left[ActiveRAF\right]}{\left(d\left(d_{rel}\right)\right)^2} = -\frac{\sqrt{1 + 6RAF_{rel}}}{324\ K_A^3 RAF_{rel}} \times \left(1 + \sqrt{1 + 6RAF_{rel}} - 2\left(1 + 6RAF_{rel}\right)\right)^2$$

This function is negative definite as long as the parameters are all positive. The latter is a given since the parameters are either concentrations (non-negative) or ratios of concentrations (equilibrium constants). The drug will show paradoxical activation via the suggested autoinhibition mechanism when the value of $d_{rel}$ is positive. We find that the maximum fold change occurs at a positive drug concentration only when the following condition is true to derive the necessary and sufficient condition for PA within this sub-model:

$$8RAF_{rel} < \left(3K_A + 1\right)\left(K_A - 1\right)$$

As the concentration ratio $RAF_{rel}$ may not be smaller than 0, the right-hand expression is always positive. Therefore, the condition that the autoinhibition equilibrium should favor the inactive RAF kinase naturally arises from the above expression (i.e., $K_A > 1$). *In other words, PA does not always happen when CA is present.* It is also clear from the same expression that a model that does not show autoinhibition ($K_A = 0$) or a model where there are no dimers ($K_{dim} \to \infty : RAF_{rel} \to 0$) will not produce PA. *Therefore, we demonstrate analytically that the presence of both CA and stabilization of the active form by RAF inhibitors is sufficient to create PA for a wide range of system parameters.*

Next, we identify the predictions of this model for the baseline RAF signaling, defined as active RAF in the absence of the drug. The expression for baseline signaling in the basic model is shown in **Supplementary file 1**. This expression is positive definite and monotonic as a function of E3, where $E3 = 8 \times RAF_{rel}/\left(1 + K_A\right)^2$. This can be observed by taking the first derivative relative to E3, which reduces to the following positive definite function:

$$\frac{d\left(baseline\_ActiveRAF\right)}{d\left(d_{rel}\right)} = \frac{\left(\sqrt{E3 + 1} - 1\right)^2}{E3^2\sqrt{E3 + 1}}$$

Therefore, baseline signaling monotonically increases as a function of RAF concentration and reduces as a function of equilibrium constant $K_A$ and equilibrium dissociation constant $K_{dim}$.

Moreover, expression for the maximum paradoxical activation can be calculated by substituting the corresponding value of drug concentration into the expression for active kinase concentration:

$$ActiveRAF|_{max} = \frac{\sqrt{6RAF_{rel} + 1} + \left(6\sqrt{6RAF_{rel} + 1} - 9\right)RAF_{rel} - 1}{27K_A RAF_{rel}}$$

Note that in this mechanism the maximal kinase activity is a simple inverse function of $K_A$ . Since the minimum value of $K_A$ to produce PA limits to 1 from above, the maximum possible PA for a kinase is a function of $RAF/K_{dim}$ only. This is also because we have modeled the autoinhibitory mechanism in the simplest possible, first-order approximation that represents the phenomena itself but may not encompass the detailed dynamics such as that modeled in the following sections. The kinase activity in the absence of the drug is shown in the following expression:

$$ActiveRAF|_{d_{rel} \to 0} = \frac{\left(-\sqrt{(K_A + 1)^2 + 8RAF_{rel}} + K_A + 1\right)^2}{8RAF_{rel}}$$

The maximal fold change (FC) in activity can therefore be calculated:

$$
\begin{aligned}
MaxFC &= \frac{ActiveRAF|_{max}}{ActiveRAF|_{d_{rel} \to 0}} \\
&= \frac{8\left(\sqrt{6RAF_{rel} + 1} + \left(6\sqrt{6RAF_{rel} + 1} - 9\right)RAF_{rel} - 1\right)}{27K_A\left(-\sqrt{(K_A + 1)^2 + 8RAF_{rel}} + K_A + 1\right)^2}
\end{aligned}
$$

When the conditions of PA are satisfied, the fold change monotonically increases as a function of the autoinhibition equilibrium constant and reduces as a function of the RAF concentration to dimerization dissociation constant ratio. Both of these facts can be derived upon evaluating the first derivative of the fold change function above and applying the constraint relationship for PA existence, which results in a positive definite derivative relative to $K_A$ and a negative definite derivative relative to $RAF_{rel}$. The implication of the former is that when the autoinhibition equilibrium favors the inactive state, the fold change relative to baseline (drug-free) condition is enhanced. Further, when the dimerization rate of RAF increases (i.e., dissociation constant $K_{dim}$ reduces), the corresponding increase in $RAF_{rel}$ leads to a reduced fold change in RAF activity in response to the drug, relative to the control.

## The relationship between unbound drug and total drug

We note here that the expressions characterizing the PA phenomenon thus far are done as a function of the free-drug concentration. This is to keep the expressions relatively simple. The total drug available in a cell also includes the drug concentration bound to RAF, including both monomeric and dimeric states of RAF. Our conclusions become final predictions of the model once we can show that unbound/free drug and the total drug concentrations are directly and monotonically related.

The relationship between total drug available (DTOT) and the unbound (free) drug ($d$) can be calculated by substituting the equilibrium conditions:

$$\frac{2[A]^2[d]([d] + K_d)}{K_d^2 K_{dim}} + \frac{[A][d]}{K_d} + [d] = Drug$$

where the unbound kinase ($A$) is a function of free drug ($d$). After splitting this function into three additive terms, we substitute the expression for $[A]$ and show that the derivative relative to unbound drug for each term is non-negative. We do not show the explicit expression for these derivatives here and refer to the Mathematica notebook provided as *Supplementary file 1*. We note that these derivatives are positive definite for all positive values of the parameters in the model. Therefore, we obtain that the total drug quantity is a monotonic increasing function of the quantity of unbound drug (*Appendix 1—figure 4*). The monotonicity of this function implies that our conclusions for paradoxical activation follow through as a function of total drug as well. However, the nonlinear relationship implies that the map between drug IC-50 values and the free-drug amount is directly related but also nonlinear. This is significant while comparing IC-50 values predicted by a modeling framework with experimental results. When fitting our RAF activity functions to experimental data, we work with the total drug concentration.

## Numerical model illustrations of three mechanisms of PA: NC, DP, and CA

*Appendix 1—figure 5* shows the mechanisms of drug potentiation of RAF dimerization (DP, blue line), NC (red dotted line), CA (green dashed line), and the unified model with all three mechanisms (unified, black dashed line). We illustrate each of the states of RAF (i–vi) for each of the sub-models, in a manner similar to what is shown in *Figure 2—figure supplement 2*, but now extended to include these additional models. The CA and DP mechanisms show distinct features in active RAF states. The DP model shows a rapid activation of RAF in response to a dimerization-inducing drug (blue, solid line) and then a rapid reduction of signaling. In contrast, the PA peak occurs at higher drug concentrations for the CA model, and the CA model also has a much wider tail of the distribution (green, dashed line). The NC model (red, dotted curves) also shows some increase of the partly

active RAF dimer at low drug concentrations. However, this mechanism alone will not generate PA as the total abundance of active RAF protomers within RAF dimers will always be less than the levels obtained in the absence of drug. This point can also be shown as a parameter-independent, analytic feature of the NC model (*Supplementary file 1*) and is in agreement with prior in silico modeling of the NC mechanism (*Kholodenko, 2015*). We show in *Figure 3—figure supplement 1A and B* that in the presence of other mechanisms that induce PA, the NC model can modulate the dose–response curve to activate RAF at higher concentrations of the drug. The nature of their contributions remains quite distinct across the DP, NC, and CA models even as they combine to create more complex dose–response.

All of the mechanisms are combined into a unified model (black, dashed), which incorporates the unique features of each of the sub-models. The curves in *Appendix 1—figure 5* identify the differences within the models and also show the synergistic combination of the mechanisms. We also note the analytic expressions for each of the models in *Supplementary file 1* are linearly independent, and therefore cannot be converted into each other by any simple translation or scaling transformation of any of the variables. The hypothesis that we could completely characterize these parameters using normalized data from signaling experiments arose from this insight into the analytic and numerical activation curves. These fits may be performed in the future for other cell lines, primary cells, mouse models, or any system of interest where such characterization may aid in drug development and precision therapy identification. The clear distinguishability of different mechanisms using fitting is an emergent feature due to the nonlinearities between how the models combine. The distinguishability is a rare feature in multiparameter models because of degeneracies that are common in multiparameter models. An example of a degeneracy in our model system involves the abundance of RAF and the dimerization equilibrium constant; these terms are difficult to isolate from each other in our fits. The coarse-grained data signaling data in our fits is much improved with narrower and more robust parameter regions predicted when the RAF concentration and dimerization rates are fixed to values based on the literature.

Finally, while NC and drug-induced increases in dimerization are behaviors that can be tuned between RAF inhibitors, we highlight that CA is a feature that spans all RAF inhibitors. This means that this process and the secondary events that potentiate this process are important factors to consider in the development of any RAF inhibitor.

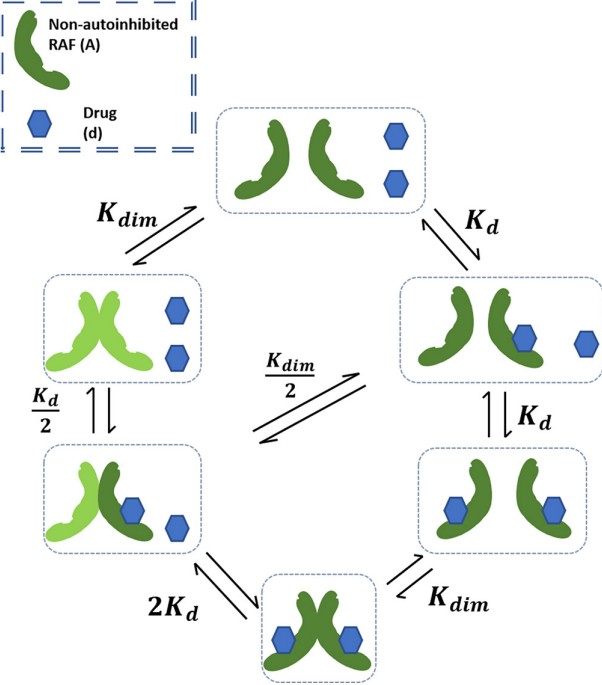

**Appendix 1—figure 1.** Schematic of base RAF dimerization and drug binding model that does not include conformational autoinhibition, negative cooperativity, or drug-induced dimerization.

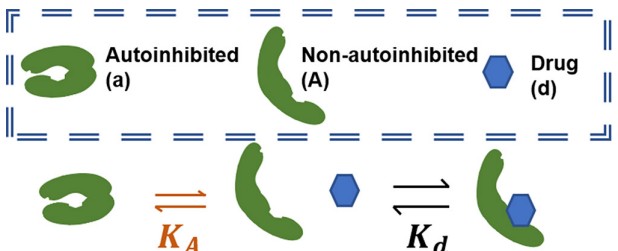

**Appendix 1—figure 2.** Schematic of conformational autoinhibition and inhibitor binding, where inhibitor can only bind the non-autoinhibited form of the target protein.

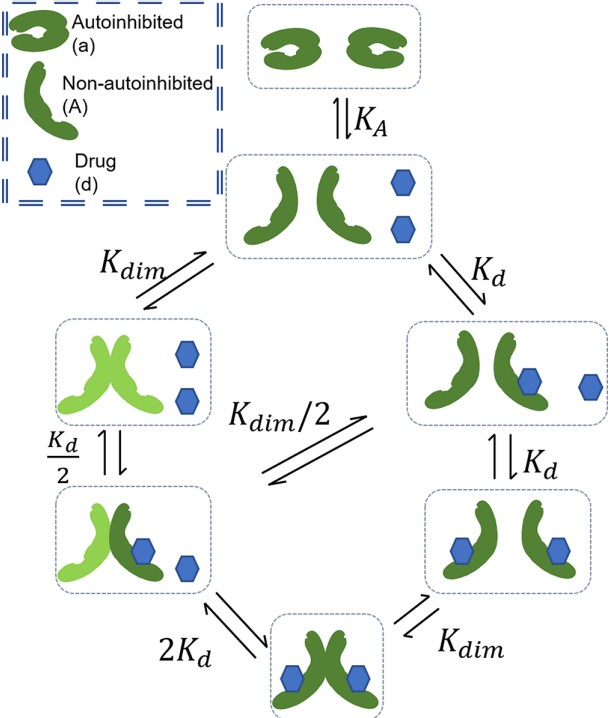

**Appendix 1—figure 3.** Schematic of the base model extended to include conformational autoinhibition.

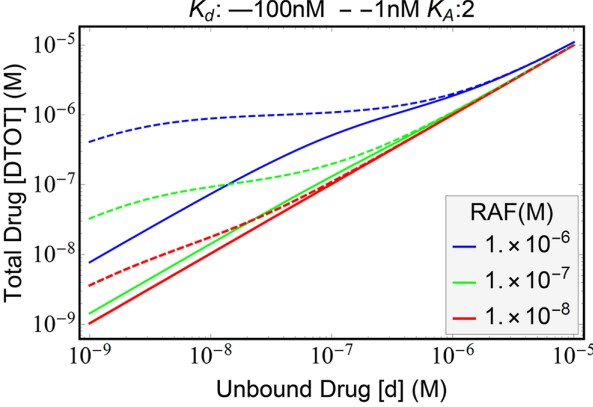

**Appendix 1—figure 4.** Plot of the relationship between unbound drug and total drug.

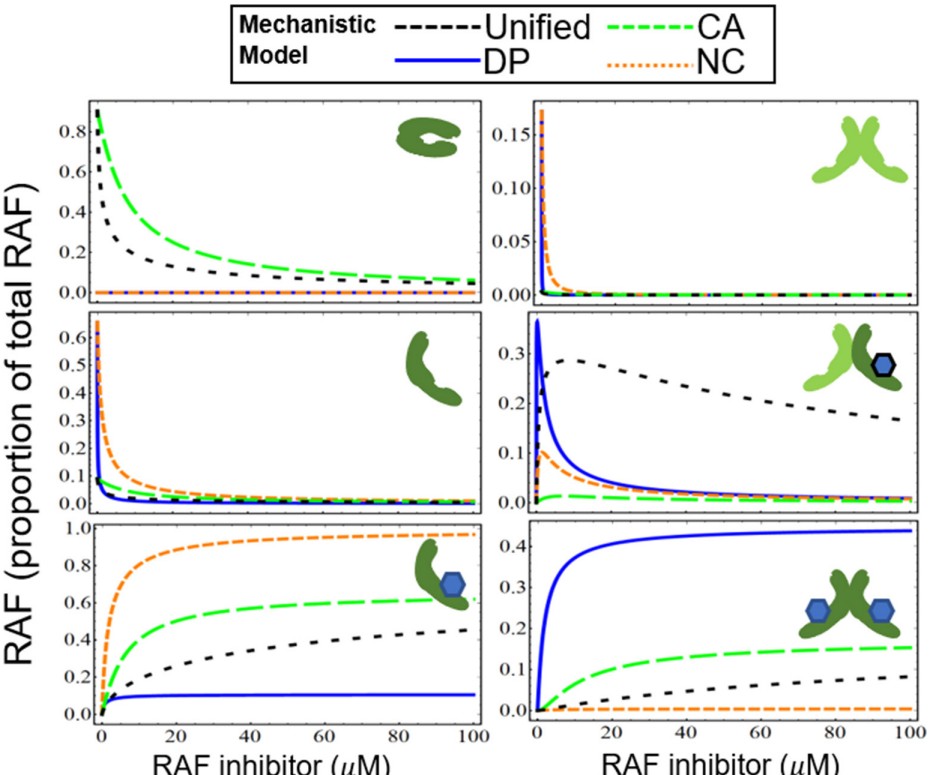

**Appendix 1—figure 5.** Visualization of how paradoxical activation (PA) arises when conformational autoinhibition with inhibitor stabilization, dimer potentiation, and/or negative cooperativity are present.

