## [Editor Report]

This important study uses mathematical modeling to demonstrate that conformational autoinhibition of the RAF kinase is an important feature of its paradoxical activation by pharmacological inhibitors. The provided theoretical analysis is highly compelling and increases the mechanistic understanding of how regulation by the N-terminal regulatory domains of RAF contributes to the paradoxical effect of the inhibitors. This article is poised to be of interest to biochemists, cell biologists, and cancer biologists, ultimately paving the way for improved RAF therapeutics.

---

## [Decision Letter]

**Decision letter after peer review:**

Thank you for submitting your article "RAF conformational autoinhibition and 14-3-3 proteins promote paradoxical activation" for consideration by *eLife*. Your article has been reviewed by 3 peer reviewers, one of whom is a member of our Board of Reviewing Editors, and the evaluation has been overseen by Volker Dötsch as the Senior Editor. The following individual involved in the review of your submission has agreed to reveal their identity: John G Albeck (Reviewer #2).

Essential revisions:

All reviewers are in agreement that the main strengths of the manuscript include a theoretical demonstration that conformational autoinhibition is an important feature of Raf kinase paradoxical activation in response to Raf-targeted therapies and the finding that paradoxical activation can arise in the absence of negative cooperativity and the dimerizing effect of Raf kinase inhibitors. There is consensus that these findings provide a significant advancement in the understanding of the Raf paradoxical activation phenomenon. Reviewers were also in agreement about the weaknesses of this study. One of them is lack of the experimental data that directly test whether paradoxical activation is reduced when conformational autoinhibition is removed. Another weakness stems from the concern that the 14-3-3-dependent effects described in the paper might not be relevant at the physiological 14-3-3 levels due to 14-3-3 abundance in a cell. Lack of the experimental data that directly test these effects is another weakness. Thus, the recommendation for publishing this work at *eLife* is pending a major revision, which could take two paths:

1) The manuscript in its current form needs to be supported by stronger direct experimental evidence. The reviewers acknowledge that experimental validation of conformational autoinhibition as a contributing factor to the paradoxical activation might be difficult and agree that the theoretical model provided is sufficient. However, all reviewers agree that the 14-3-3 regulation should be supported by more direct experimental evidence that eliminates the ambiguity of interpretation that stems from conditions of 14-3-3 overexpression. A recommendation for such an experiment is provided by Reviewer #1.

2) Acknowledging that experimental work might be out of the scope of the PI's lab expertise, another proposed revision path is to remove the section that describes the 14-3-3-based regulation from the current manuscript. All reviewers agree that this would make a more focused paper that would highlight its most impactful findings and eliminate concerns about how 14-3-3 levels in a cell can be regulated to functionally contribute to the proposed effects.

*Reviewer #1 (Recommendations for the authors):*

1. The reported roles of conformational autoinhibition and 14-3-3 binding in PA of RAF are very exciting findings of this study but remain to be experimentally validated. The phosphorylation site on RAF, which recruits 14-3-3 stabilizing RAF autoinhibition is known, and mutating this site should destabilize the autoinhibited state of RAF. This study would be greatly enhanced by the analysis of such, or functionally analogous mutants that disrupt RAF inactive state, to investigate the contribution of conformational autoinhibition of RAF to PA.

2. To evaluate the contribution of 14-3-3 to PA of RAF, authors investigate the roles of 14-3-3 in conformational autoinhibition stabilization (CAS) or dimer stabilization (DS) models (Figure 4C). By doing so they found DS failed to potentiate PA. The authors do not elaborate much on this, but since DS is an established mechanism to potentiate PA, why do the authors think DS does not play a role in this context? Could the authors use RAF and/or 14-3-3 mutations to decouple the CAS and DS effect of 14-3-3?

3. Figure 1. title sound overreaching as the figure does not show an analysis of PA with mathematical models.

4. Figure 1A. graph axes should indicate which way RAF inhibitor concentration and RAF activity increase for X and Y axes, respectively.

5. Figure 4C typing error, CAS+MS should be CAS+DS.

*Reviewer #2 (Recommendations for the authors):*

Overall my recommendations mainly focus on clarifying how some of the model results are presented and explained, as the current representation isn't fully adequate to see how the fits are working.

1. It should be considered whether a direct test of the role of conformational autoinhibition can be performed using Raf proteins that lack the autoinhibitory domain, which has been used for some time. Understandably, such an experiment could be difficult to execute, as it would likely require the other native forms of Raf to be deleted. Alternatively, it may be the case that data in the literature using such mutants could provide insight. If neither of these is possible, it would be helpful to at least discuss how such an experimental test could be set up and how its outcome could support or contradict the model.

2. It is somewhat confusing what is going on with the fitted parameter sets in Figure 3. The text refers to 28 or 30 parameter sets, and it is not clear what these parameters are or which two parameters are omitted in the 28. How do the 28 parameters relate to the 27 drug-specific parameters in Table S1? Also, what do the different iterations of the same symbol represent in Figure 3C and Figure S2C? Overall, in addition to clarifying these ambiguities, it would be helpful to present the complete parameter sets more comprehensively, such as in a heatmap, in addition to the f vs. Kd plot.

3. The discussion of the fits of the different model variants makes the statement that "Our analysis also suggests that all three mechanisms make important contributions to the overall PA behavior in a drug-specific manner", but the drug-specific aspect of these fits is not detailed in any of the figures. This should be made more clear by showing the fits for specific types of inhibitors for each of the models, to support this statement.

4. The "clustering" of different inhibitor types in the f vs. Kd plot is a somewhat loose interpretation given the sparseness of the points within the space and their relative distances. It seems like it would be more accurate to say that inhibitors of the same type occupy common regions of parameter space. Also, given that g values also differ over at least 10-fold, it would be helpful to include a visualization that shows how this factor distributes as well.

*Reviewer #3 (Recommendations for the authors):*

First, a comment on the scope of my review. I approach the paper with a deep appreciation for the power of mathematical modeling and an abiding interest in RAF regulation and the PA phenomenon, but I am not a mathematician. Thus, I leave review of this aspect of the paper to other referees and simply assume that the authors have gotten the math right – that they have properly implemented the models as they've described them.

A couple of things you might consider:

1. At a fixed 14-3-3 concentration, I would think that the effect of 14-3-3 binding could be subsumed with adjustments to Ka and Kdim. If this is correct, it would be helpful to point it out and if not, to explain why not.

2. You nicely show how PA varies with Ka in the model in Fig. 2. It would be helpful to also see how it varies with Kd and Kdim. Are there regimes of inhibitor affinity in which is it more/less pronounced?

---

## [Author Response]

Essential revisions:All reviewers are in agreement that the main strengths of the manuscript include a theoretical demonstration that conformational autoinhibition is an important feature of Raf kinase paradoxical activation in response to Raf-targeted therapies and the finding that paradoxical activation can arise in the absence of negative cooperativity and the dimerizing effect of Raf kinase inhibitors. There is consensus that these findings provide a significant advancement in the understanding of the Raf paradoxical activation phenomenon. Reviewers were also in agreement about the weaknesses of this study. One of them is lack of the experimental data that directly test whether paradoxical activation is reduced when conformational autoinhibition is removed. Another weakness stems from the concern that the 14-3-3-dependent effects described in the paper might not be relevant at the physiological 14-3-3 levels due to 14-3-3 abundance in a cell. Lack of the experimental data that directly test these effects is another weakness. Thus, the recommendation for publishing this work at eLife is pending a major revision, which could take two paths:1) The manuscript in its current form needs to be supported by stronger direct experimental evidence. The reviewers acknowledge that experimental validation of conformational autoinhibition as a contributing factor to the paradoxical activation might be difficult and agree that the theoretical model provided is sufficient. However, all reviewers agree that the 14-3-3 regulation should be supported by more direct experimental evidence that eliminates the ambiguity of interpretation that stems from conditions of 14-3-3 overexpression. A recommendation for such an experiment is provided by Reviewer #1.2) Acknowledging that experimental work might be out of the scope of the PI's lab expertise, another proposed revision path is to remove the section that describes the 14-3-3-based regulation from the current manuscript. All reviewers agree that this would make a more focused paper that would highlight its most impactful findings and eliminate concerns about how 14-3-3 levels in a cell can be regulated to functionally contribute to the proposed effects.

We thank the reviewers and the editor for the reviews and for the options. We are pleased that there is agreement about the significant value of recognizing the conformational autoinhibition mechanism as an essential part of understanding paradoxical activation.

We are resubmitting under option #2, the proposed option to focus on the theoretical results and to eliminate the section on 14-3-3. We are very pleased with the agreement about the value of the theoretical work. Our revised manuscript is focused, as the reviewers stated it would be, and we have made changes and additions to address the reviewer’s requests. The requests were all very logical and these additions further strengthen the manuscript. We are very pleased with how this has turned out, and we hope the reviewers and editors are also happy with the revision.

Reviewer #1 (Recommendations for the authors):1. The reported roles of conformational autoinhibition and 14-3-3 binding in PA of RAF are very exciting findings of this study but remain to be experimentally validated. The phosphorylation site on RAF, which recruits 14-3-3 stabilizing RAF autoinhibition is known, and mutating this site should destabilize the autoinhibited state of RAF. This study would be greatly enhanced by the analysis of such, or functionally analogous mutants that disrupt RAF inactive state, to investigate the contribution of conformational autoinhibition of RAF to PA.

Although we have removed the 14-3-3 section and are not doing experiments in the present study, we agree that this would be an excellent approach for experimental testing of the theoretical ideas presented in the manuscript. We have extended our discussion to discuss this phosphorylation event and mutations that disrupt it, and how they may be useful for future experimental work. Specifically, we have updated the manuscript’s discussion to state:

“Experimental testing of model predictions is an important next step. One possible approach for experimentally investigating the role of conformational autoinhibition on PA would involve RAF proteins that are defective in conformational autoinhibition. Of note, there are some RAF mutants in Noonan syndrome that are believed to be activating due to impaired regulation of conformational autoinhibition. For example, stabilization of the autoinhibited form of CRAF involves the phosphorylation of serine at residue 259 followed by 14-3-3 binding to the phosphorylated serine. The S257L, S259F, P261A, and N262K CRAF mutations are impaired at binding to 14-3-3 and have severely impaired phosphorylation of serine 259 (Kobayashi *et al.,* 2010; Pandit *et al.,* 2007). Alternatively, truncation mutants lacking the autoinhibitory domain could be used. In such experiments, comparisons of RAF inhibitor dose responses between cells expressing one such mutant against control cells expressing the wild-type equivalent RAF protein would reveal whether there were changes in PA fold-change and PA range (as defined in Figure 2). We would expect to see reduced PA fold-change and reduced PA range for the conformational autoinhibition impaired mutants. We anticipate PA range would be the more useful observable because nonlinearities in downstream signaling (including possible saturation of readouts like ERK phosphorylation) may make measurements of fold-change less useful. As PA range is determined by when the signal passes back through the baseline level after the increase induced by a RAF inhibitor, it should be less impacted by nonlinear signal processing downstream from RAF. Whether the experiment utilizes Noonan syndrome RAF1 mutants or truncated RAF mutants, the presence of endogenous, wild-type BRAF, ARAF, and CRAF may make it difficult to observe differences in PA due to an introduced mutant, so such experiments may need to be performed in cells where the three endogenous RAF genes are knocked out or silenced.”

2. To evaluate the contribution of 14-3-3 to PA of RAF, authors investigate the roles of 14-3-3 in conformational autoinhibition stabilization (CAS) or dimer stabilization (DS) models (Figure 4C). By doing so they found DS failed to potentiate PA. The authors do not elaborate much on this, but since DS is an established mechanism to potentiate PA, why do the authors think DS does not play a role in this context? Could the authors use RAF and/or 14-3-3 mutations to decouple the CAS and DS effect of 14-3-3?

In the revision, we have cut this section per Option 2 of focusing on conformational autoinhibition. In our future work, we plan to pursue experiments that will use mutants to focus on just one role of 14-3-3 on RAF, in line with the suggestion.

3. Figure 1. title sound overreaching as the figure does not show an analysis of PA with mathematical models.

We have updated the title of Figure 1, thank you for catching this.

4. Figure 1A. graph axes should indicate which way RAF inhibitor concentration and RAF activity increase for X and Y axes, respectively.

We have updated our schematic in Figure 1A to indicate which direction has increasing RAF inhibitor and increasing RAF activity (x and y axes, respectively) and noted this in our updated figure legend. Thank you for this suggestion.

5. Figure 4C typing error, CAS+MS should be CAS+DS.

Thank you for catching this. Although we have now removed the 14-3-3 section that included this, we have made the correction so that we will not make the same mistake in our continuing studies. Thank you.

Reviewer #2 (Recommendations for the authors):Overall my recommendations mainly focus on clarifying how some of the model results are presented and explained, as the current representation isn't fully adequate to see how the fits are working.1. It should be considered whether a direct test of the role of conformational autoinhibition can be performed using Raf proteins that lack the autoinhibitory domain, which has been used for some time. Understandably, such an experiment could be difficult to execute, as it would likely require the other native forms of Raf to be deleted. Alternatively, it may be the case that data in the literature using such mutants could provide insight. If neither of these is possible, it would be helpful to at least discuss how such an experimental test could be set up and how its outcome could support or contradict the model.

We have added a section in the discussion that elaborates on some experimental testing. It is copy-and-pasted in the response to Reviewer #1. This new section mentions the potential for truncation mutants without the autoinhibitory domain, or for Noonan syndrome RAF1 mutants with impaired autoinhbition (due to impaired 14-3-3 stabilization of the closed form) to serve as possible vehicles to test the role of CA. The challenge of the endogenous, native forms is also mentioned. We also discuss what we expect (reduced PA with these mutants) and which read-out (PA range) should be the better measurement and why. Thank you for this suggestion. We think this addition should make the theoretical results of greater interest to experimental biologists, and we appreciate the suggestion. Thank you.

2. It is somewhat confusing what is going on with the fitted parameter sets in Figure 3. The text refers to 28 or 30 parameter sets, and it is not clear what these parameters are or which two parameters are omitted in the 28. How do the 28 parameters relate to the 27 drug-specific parameters in Table S1?

Thank you for requesting us to clarify this section. We agree that this the previous version was confusing and should have been more clearly communicated. We have revised and updated the section where these parameters are introduced to be more clear and more thorough. Specifically, the new text reads:

“We set values for two key parameters, RAF abundance and RAF dimerization equilibrium constant based on estimates and do not fit them. A table listing the best-fit parameters of each of the sub-models, are indicated in Supplementary File 1. Of note, we have one global parameter (the autoinhibition equilibrium constant) and twenty-seven drug-specific parameters. Our twenty eight total parameter estimates yield a model that matches the experimental data well (Figure 3B). Details of our approach for identifying model parameters are provided in the methods, and our code is in the supplement. We obtain similar quality overall fits if we do not pre-specify the RAF abundance and the RAF dimerization constant and add these two parameters to the other twenty eight parameters and then perform the same parameter estimation procedure (Figure 3 —figure supplement 1C-E). We observed that some parameters could be constrained to a narrow region through this procedure, while other parameters could vary widely and still match the same data (Figure 3 —figure supplement 2,3).”

Also, what do the different iterations of the same symbol represent in Figure 3C and Figure S2C?

We have updated the text to clarify this in multiple places. First, within the main text, where the parameter fits are first mentioned we added the following sentence:

“We developed a computational process for obtaining sets of parameters that fit our model to published experimental data (Karoulia et al., 2016). Our approach finds multiple parameter sets for the same drug that are within 10% of the error for the parameter set with the least error. Within the text, we also have the following sentence to communicate the value of having multiple, similar quality, fits.”

We also plot alternative parameter estimates that are within 10% error of the considered parameter set with the least error, and this helps demonstrate the robustness of this observation.

Within the figure legend, we have added this sentence:

“For each drug, we show all obtained best fit parameter sets that were within 10% of best fit metric.”

Overall, in addition to clarifying these ambiguities, it would be helpful to present the complete parameter sets more comprehensively, such as in a heatmap, in addition to the f vs. Kd plot.

We have made three changes to more thoroughly communicate the parameter sets.

– We have added a supplementary figure (Figure 3 —figure supplement 3) that shows the values and the ranges for the 28 parameters (3 parameters x 9 drugs, and one global KA parameters), and we show these for all variants of the model (full model, and versions where one or more of the three different mechanisms of the model are systematically excluded).

– We have added a supplementary figure (Figure 3 —figure supplement 2) that shows the best fits for the model and different submodels analyzed in Figure 3.

– We have extended and improved our supplementary table to present the best fit parameters for all sub-models. (Supplementary File 1)

3. The discussion of the fits of the different model variants makes the statement that "Our analysis also suggests that all three mechanisms make important contributions to the overall PA behavior in a drug-specific manner", but the drug-specific aspect of these fits is not detailed in any of the figures. This should be made more clear by showing the fits for specific types of inhibitors for each of the models, to support this statement.

We have elaborated and expanded this section. The new text now reads:

“Altogether, we believe that our analysis provides strong support for conformational autoinhibition being a critical factor to PA. Our sub-model analysis suggests that both conformational autoinhibition and dimer potentiation are necessary for the mathematical model to reproduce the experimental data with the smallest degree of error. Even though the negative cooperativity term ‘g’ was not strongly constrained, and even though it had the smallest impact on the ability of the model to fit the experimental data, the model fits consistently had values of g that were much larger than one (Table S1) which is consistent with negative cooperativity. Thus, our analysis suggests that all three processes contribute to the overall PA phenomenon.”

4. The "clustering" of different inhibitor types in the f vs. Kd plot is a somewhat loose interpretation given the sparseness of the points within the space and their relative distances. It seems like it would be more accurate to say that inhibitors of the same type occupy common regions of parameter space.

We have made the following changes:

– Eliminated the use of the term “clustering”.

– Rephrased that section to read:

“Intriguingly, when we considered the collection of best-fit parameters we found the different RAF inhibitors appeared to separate in a manner reflective of their known identity as Type I, Type I.5, and Type II inhibitors (Figure 3C, Supplementary File 1).”

Also, given that g values also differ over at least 10-fold, it would be helpful to include a visualization that shows how this factor distributes as well.

We have added a new supplementary figure (Figure 3 —figure supplement 3), which includes box-and-whisker plots that show the range of parameter values for g for each of the 9 drugs fit to, as well as f, Kd, and KA values. Thank you for your review and the helpful suggestions to improve the manuscript.

Reviewer #3 (Recommendations for the authors):First, a comment on the scope of my review. I approach the paper with a deep appreciation for the power of mathematical modeling and an abiding interest in RAF regulation and the PA phenomenon, but I am not a mathematician. Thus, I leave review of this aspect of the paper to other referees and simply assume that the authors have gotten the math right – that they have properly implemented the models as they've described them.A couple of things you might consider:1. At a fixed 14-3-3 concentration, I would think that the effect of 14-3-3 binding could be subsumed with adjustments to Ka and Kdim. If this is correct, it would be helpful to point it out and if not, to explain why not.

We have added the following sentences to the manuscript:

“Intuitively, stabilization of RAF conformational autoinhibition by itself would result in a stronger PA and could be approximated as an increase in the K_A_ term within our model. RAF stabilization of RAF dimers could be approximated as in increase in the K_dim_ term within our model. Thank you for this suggestion.”

2. You nicely show how PA varies with Ka in the model in Fig. 2. It would be helpful to also see how it varies with Kd and Kdim. Are there regimes of inhibitor affinity in which is it more/less pronounced?

We have added new supplementary figure (Figure 2 —figure supplement 1) that show this information.